∂ | **Open Peer Review** | Host-Microbial Interactions | Research Article

# Impact of the elderly lung mucosa on *Mycobacterium tuberculosis* transcriptional adaptation during infection of alveolar epithelial cells

Angélica M. Olmo-Fontánez,[1,2] Anna Allué-Guardia,[1,3] Andreu Garcia-Vilanova,[1] Jeremy Glenn,[1] Shu-Hua Wang,[4] Robert E. Merritt,[5] Larry S. Schlesinger,[1,3] Joanne Turner,[1,3] Yufeng Wang,[6] Jordi B. Torrelles[1,3]

**ABSTRACT** Tuberculosis is one of the leading causes of death due to a single infectious agent. Upon infection, *Mycobacterium tuberculosis* (*M.tb*) is deposited in the alveoli and encounters the lung mucosa or alveolar lining fluid (ALF). We previously showed that, as we age, ALF presents a higher degree of oxidation and inflammatory mediators, which favors *M.tb* replication in human macrophages and alveolar epithelial cells (ATs). Here, we define the transcriptional profile of *M.tb* when exposed to healthy ALF from adult (A-ALF) or elderly (E-ALF) humans before and during infection of ATs. Prior to infection, *M.tb* exposure to E-ALF upregulated genes essential for bacterial host adaptation directly involved in *M.tb* pathogenesis. During infection of ATs, E-ALF exposed *M.tb* further upregulated genes involved in its ability to escape into the AT cytosol bypassing critical host defense mechanisms, as well as genes associated with defense against oxidative stress. These findings demonstrate how alterations in human ALF during the aging process can impact the metabolic status of *M.tb*, potentially enabling a greater adaptation and survival within host cells. Importantly, we present the first transcriptomic analysis on the impact of the elderly lung mucosa on *M.tb* pathogenesis during intracellular replication in ATs.

**IMPORTANCE** Tuberculosis is one of the leading causes of death due to a single infectious agent. Upon infection, *Mycobacterium tuberculosis* (*M.tb*) is deposited in the alveoli and comes in contact with the alveolar lining fluid (ALF). We previously showed that elderly ALF favors *M.tb* replication in human macrophages and alveolar epithelial cells (ATs). Here we define the transcriptional profile of when exposed to healthy ALF from adult (A-ALF) or elderly (E-ALF) humans before and during infection of ATs. Prior to infection, exposure to E-ALF upregulates genes essential for bacterial host adaptation and pathogenesis. During infection of ATs, E-ALF further upregulates *M.tb* genes involved in its ability to escape into the AT cytosol, as well as genes for defense against oxidative stress. These findings demonstrate how alterations in human ALF during the aging process can impact the metabolic status of *M.tb*, potentially enabling a greater adaptation and survival within host cells.

**KEYWORDS** *Mycobacterium tuberculosis*, aging, alveolar epithelial cells, alveolar lining fluid, transcriptomics

**Peer Reviewer** Rajko Reljic, St George's University of London, London, United Kingdom

Address correspondence to Angélica M. Olmo-Fontánez, aolmo@txbiomed.org, Anna Allué-Guardia, aallueguardia@txbiomed.org, or Jordi B. Torrelles, jtorrelles@txbiomed.org.

The authors declare no conflict of interest.

The increase in the worldwide elderly population (60 years and older) is projected to double to 2.1 billion by 2050 (1), and close to 95 million by 2060 in the United States alone (2). Unfortunately, living longer is associated with ongoing molecular and physiological changes of natural lung aging, leading to decreased lung function and weakened immune responses, increasing susceptibility to lung diseases and respiratory

infections (3–7). Consequently, the elderly population is at high risk of contracting respiratory diseases, including tuberculosis (TB) (5, 8).

More than 4,300 people die from TB daily, making it a top infectious killer globally (9). TB is caused by the airborne pathogen *Mycobacterium tuberculosis* (*M.tb*), spread mainly through inhalation and deposited into the distal portion of the airways and alveoli. In this environment, *M.tb* comes in contact with the lung mucosa or alveolar lining fluid (ALF), which contains soluble innate factors such as surfactant proteins A and D (SP-A/SP-D), complement, hydrolytic enzymes, antimicrobial peptides, and others, which can trigger subsequent innate and adaptive immune responses (10, 11). The increased susceptibility of the elderly to TB may be attributed to age-related alterations in the lung environment, including innate soluble components of the ALF. Our research has shown that aging impacts the pulmonary environment by causing a shift toward a pro-inflammatory and pro-oxidative state, thus reducing the functionality of soluble innate immune proteins such as SP-A and -D (12–14). Age-associated alterations in ALF components and their functional status are linked to decreased ability of phagocytes to control *M.tb* infection *in vitro* and *in vivo* (13, 15).

Alveolar epithelial type cells (ATs) are non-professional phagocytes that line the alveolar epithelium forming a physical barrier, preventing microbial invasion, and directly contributing to host defense against *M.tb* infection (11, 16, 17). ATs consist of two distinct cell types; type I (ATIs), the most prevalent cell type that gives the alveolus its structured shape and permits gas exchange (17), and type II (ATIIs), which play an essential role in maintaining alveolar homeostasis (11, 18). Our previous studies showed that *M.tb* exposure to elderly ALF promotes increased *M.tb* replication and growth within ATs, linked to altered intracellular trafficking and resulting in bacterial translocation into the ATs cytosol (15), dampening host protective immune responses.

Here, we first aimed to define *M.tb* metabolic adaptations upon exposure to different human ALF environments [adult (A)-ALF: 18–45 years of age; and elderly (E)-ALF: >69 years of age] prior to infection. Then, we determined the transcriptional status of A- and E-ALF-exposed *M.tb* during infection in ATs. We and others have shown *M.tb*'s ability to escape into the host cytosol (15, 19–23), hence we particularly focused on genes related to the ESX system and phthiocerol dimycoserosate (PDIM) production as potential strategies of bacterial translocation, as well as other genes associated with *M.tb* pathogenicity. Our findings indicate that exposure of *M.tb* to E-ALF alters its transcriptional profile prior to infection, but also enhances the expression of virulence-associated genes (mainly the ESX secretion system and oxidative stress defense mechanisms) during infection, allowing *M.tb* to better adapt for survival within host cells.

## RESULTS

### Differential expression analysis of ALF-exposed *M.tb* prior to infection

Our previous studies demonstrated that when *M.tb* contacts the human lung mucosa, ALF hydrolytic enzymes modify the *M.tb* cell envelope, eliciting innate and adaptive immune responses that impact subsequent *M.tb* infection outcomes *in vitro* and *in vivo* (10, 13, 24). We recently reported that exposure to E-ALF drives *M.tb*'s translocation into the cytosol, favoring its replication and growth in ATs (15). Based on these findings, we explored the transcriptional changes that *M.tb* experiences after being exposed to human A-ALF or E-ALF at their physiological concentration within the lung. We focused on the following mechanisms reported to play a role in *M.tb* escape into the cytosol: (i) ESX secretion system, (ii) phospholipases, (iii) PDIMs biosynthesis, (iv) others, including pathways reported to play a role in cytosol translocation of other pathogens. Experimental conditions were designated as A-ALF-exposed *M.tb* (A, pool of $n = 4$ different human donors), E-ALF-exposed *M.tb* (E, pool of $n = 4$ different human donors), and unexposed *M.tb* (UE) (Fig. 1A; Fig. S1). To determine the effects of ALF exposure on *M.tb* prior to infection, we performed differential expression (DE) analysis [Log$_2$ FC equal or greater than an absolute value of 1 and false discovery rate (FDR) < 0.1] of ALF-exposed *M.tb*

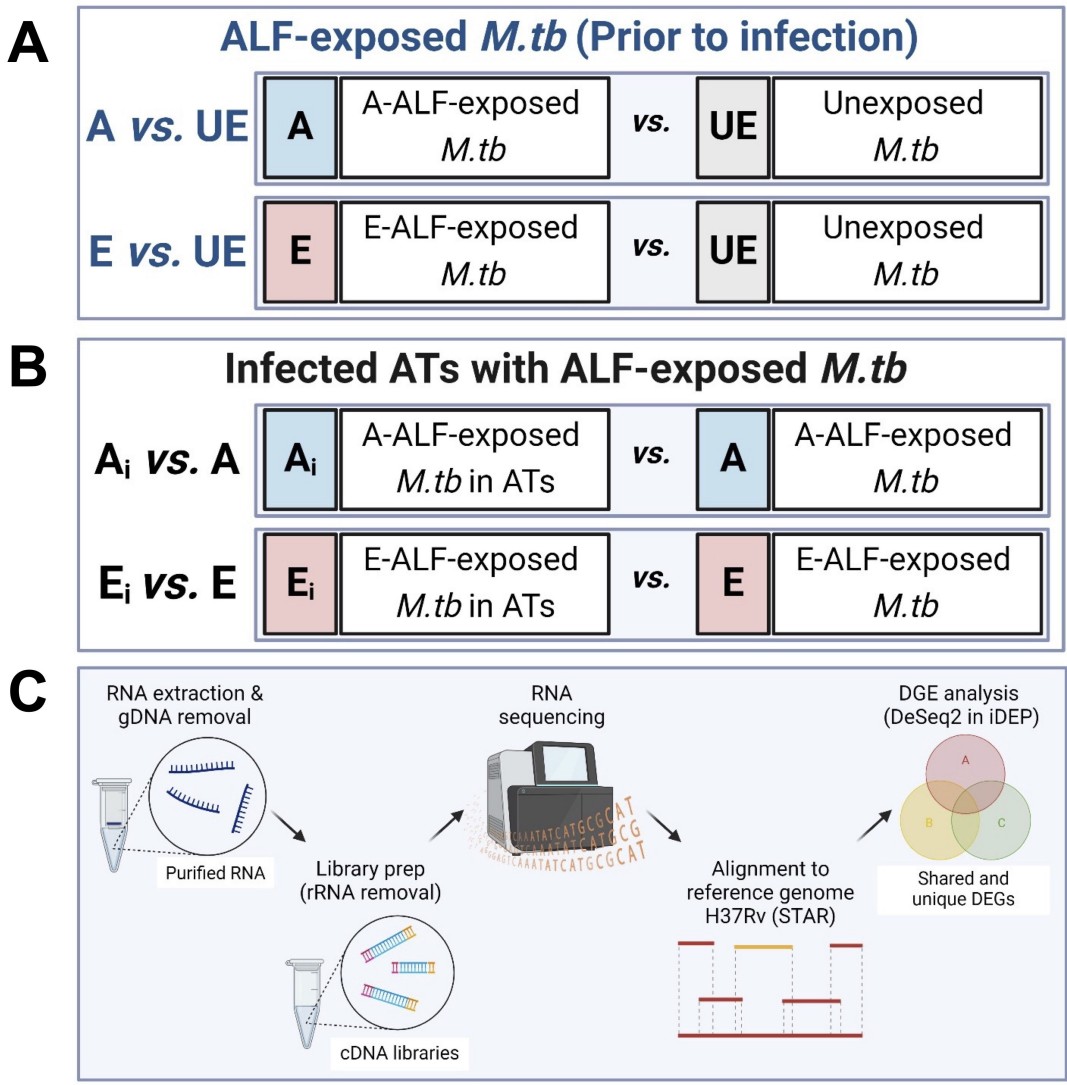

**FIG 1** Illustration of the experimental conditions and data comparisons performed in this study. (A) Prior to and (B) during AT infection. (C) Experimental strategy for RNA-seq and data analyses. Figure created using BioRender (https://biorender.com/).

(A-ALF or E-ALF) compared to unexposed *M.tb* (A vs UE and E vs UE comparisons, respectively, Fig. 1A and C).

Compared to unexposed *M.tb* (UE), A-ALF-exposed *M.tb* had 46 differentially expressed genes (DEGs) (29 upregulated and 17 downregulated) (A vs UE, Fig. 2A), while E-ALF-exposed vs unexposed *M.tb* had a total of 131 DEGs (116 upregulated and 15 downregulated) (E vs UE, Fig. 2A). Volcano plots with the top 10 DEGs are shown in Fig. S2A. Twenty-three DEGs were shared between the two (Fig. 2B; Fig. S3A). DEGs were further sorted into functional categories (Fig. 2C and D; Tables S1 and S2). In A vs UE, most DEGs associated with "Lipid metabolism" (6/7), "Conserved hypotheticals" (6/7), and "Information pathways" (1/1) were upregulated, whereas most DEGs linked to "Cell wall and cell processes" (5/7) and "Regulatory proteins" (2/3) were downregulated (Fig. 2C). Like A vs UE, E vs UE showed upregulation of most DEGs associated with "Conserved hypotheticals" (32/34), "Information pathways" (3/3), and "Lipid metabolism" (9/9), as well as downregulation of "Regulatory proteins" (3/4) (Fig. 2D). However, contrary to A vs

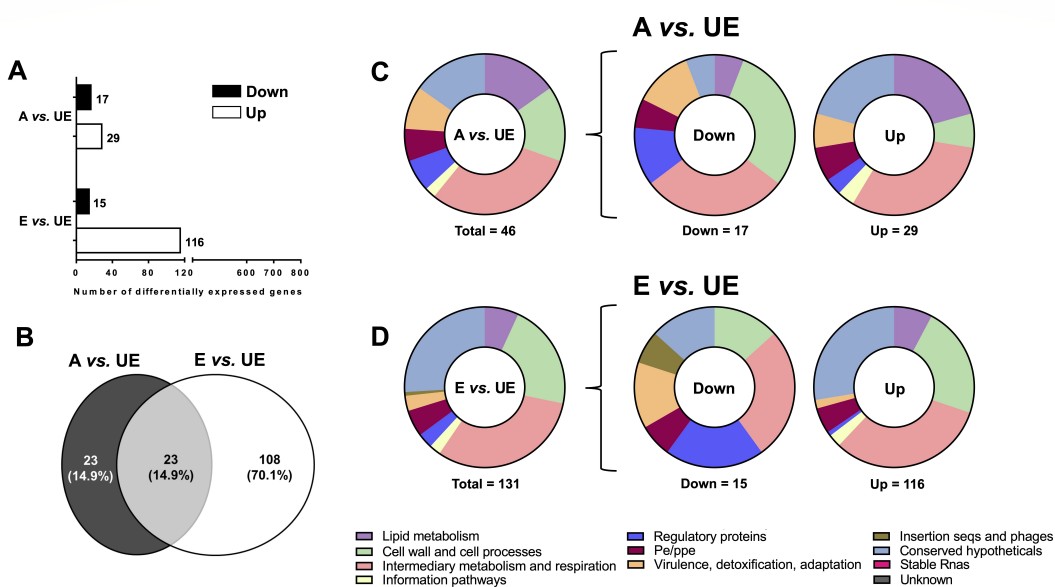

**FIG 2** Differential expression analysis of ALF-exposed *M.tb* prior to AT infection. (A) The number of differentially expressed genes in ALF-exposed *M.tb* vs unexposed *M.tb* prior to infection obtained with the DESeq2 method in the iDEP software, with the following settings: Log$_2$ FC equal or greater than an absolute value of 1, FDR < 0.1. Upregulated genes are shown in white bars and downregulated genes in black bars. (B) Venn diagram of DEGs of ALF-exposed *M.tb* vs unexposed *M.tb* prior to infection showing unique and shared DEGs in ALF-exposed *M.tb* prior to ATs infection. (C) Total number of DEGs distributed by functional categories for A-ALF-exposed vs unexposed *M.tb* (A vs UE) and (D) for E-ALF exposed vs Unexposed *M.tb* (E vs UE) prior to ATs infection (represented as a percentage). Graphs were generated in GraphPad Prism v9.1.1. A, A-ALF-exposed *M.tb*; E, E-ALF-exposed *M.tb*; UE, unexposed *M.tb*.

UE, in E vs UE most "Cell wall and cell processes" (26/28) DEGs were upregulated, while "Insertion seqs and phages" (1/1) DEGs were downregulated (Fig. 2D).

## E-ALF-exposed *M.tb* upregulates genes associated with ESX-secretion systems and phospholipases

*M.tb* has acquired specialized protein transport machinery including the type VII secretion systems (T7SSs) or ESX systems (25), composed of five distinct mycobacterial T7SSs (ESX-1 to ESX-5), with significant roles in virulence, such as nutrient uptake, immune evasion (e.g., phagosomal escape), immune modulation, horizontal gene transfer, and cell physiology (25, 26). Thus, we evaluated DEGs associated with all five ESX secretion systems (Fig. 3) as a potential mechanism associated with *M.tb* translocation into the cytosol (21, 27–29). Although expression of the majority of the genes related to the ESX-1, ESX-2, and ESX-3 secretion systems was not significantly altered in either A or E vs UE, E vs UE exhibited a higher (Log$_2$ FC > 1) but not statistically significant (FDR > 0.1) expression of ESX-1 genes *Rv3904C* (*esxE*) and *Rv3903C* (*cpnT*) (Fig. 3A), reported to mediate phagosomal disruption in *M.tb* (29, 30). In addition, phagosomal rupture and translocation of *Mycobacterium abscessus* into the host cytosol is facilitated by the ESX-4 secretion system (27), which substitutes for the ESX-1 secretion system in *M. abscessus* (27, 31). Interestingly, our results indicate that only the exposure to E-ALF resulted in a significant upregulation of DEGs (*eccC4*, *eccD4*, and *mycP4*) associated with the ESX-4 secretion system (Fig. 3B). While ESX-1, ESX-2, and ESX-4 secretion systems are essential for *M.tb* growth, the ESX-3 and ESX-5 have been related to immune modulation (26). In this regard, we found transcriptional changes in some PE/PPE genes associated with the ESX-5 system such as *PPE18* (significantly upregulated in A vs UE) and *PPE65* (significantly upregulated in E vs UE), and their functions are attributed to immunological regulation (32) (Fig. 3C).

We also aimed to describe how exposure to human ALF modulates additional virulence factors involved in *M.tb* pathogenesis. Interestingly, like the production of

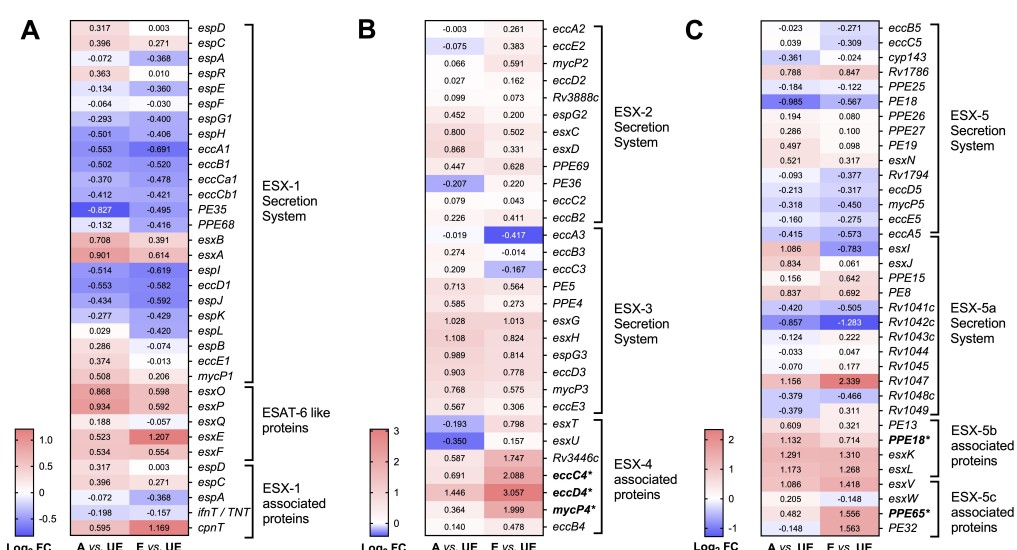

**FIG 3** Heatmaps of *M.tb* genes associated with ESX-secretion systems prior to AT infection. (A) ESX-1 and ESAT-6-like proteins; (B) ESX-2, ESX-3, and ESX-4; and (C) ESX-5 secretion system after A- or E-ALF exposure, prior to infection. Cells depict $Log_2$ FC values in ALF-exposed *M.tb* (A, Adult ALF; E, Elderly ALF) vs unexposed bacteria (UE), upregulated: red, downregulated: blue. Genes in bold indicate significant DEGs ($Log_2$ FC equal or greater than an absolute value of 1, and FDR < 0.1) for both conditions. Genes with an asterisk indicate significance in only one of the comparisons, highlighting differences between conditions. Notice the different scales used in A, B, and C, for better visualization of the results. Heatmaps were generated in GraphPad Prism v9.1.1.

phospholipases by *Listeria* to escape into the cytosol (33, 34), we found that phospholipase-related genes *Rv2349C* (*plcC*), *Rv2350C* (*plcB*), and *Rv2351C* (*plcA*) were uniquely upregulated in E vs UE (Table 1). In addition, a few other virulence genes that encode phosphatases and proteases were only upregulated in E vs UE, including *Rv2577*, *Rv1983* (*PE_PGRS35*), and *Rv3090* (Table 1). These are described to play a role in *M.tb* pathogenicity by promoting phagosomal arrest, increased replication, and dissemination (35, 36).

## Differential expression analysis of ALF-exposed *M.tb* during AT infection

We reported that *M.tb* exposed to human E-ALF replicates faster in human macrophages and ATs *in vitro* and in the lungs of *mice in vivo* (13, 15). Furthermore, in ATs, we found that at 72 h, *M.tb* translocates from endosomal compartments into the cytosol, conferring an advantageous niche for faster *M.tb* replication and survival within ATs (15). Thus, herein we determined the transcriptional changes of *M.tb* exposed to A-ALF ($A_i$) or E-ALF ($E_i$) during AT infection compared to ALF-exposed *M.tb* prior to infection (A or E) ($A_i$ vs A, and $E_i$ vs E comparisons, respectively, Fig. 1B and C; Fig. S1). According to the DE analysis, both comparisons had a similar number of upregulated and downregulated genes (Fig. 4A) and shared 1,059 DEGs (Fig. 4B; Fig. S3B). Volcano plots with the top 10 DEGs for each comparison are shown in Fig. S2B. Comparable numbers of DEGs in each functional category were found in both (Fig. 4C and D; Tables S1 and S2). Therefore, the functional category distribution did not differ remarkably between $A_i$ vs A and $E_i$ vs E (Fig. 4C and D).

## Effects of A- vs E-ALF exposure on the expression of *M.tb* ESX-associated genes during AT infection

Our results indicate that when *M.tb* is exposed to either A-ALF or E-ALF, most of the ESX-1-related genes, including those encoding for ESAT-6 (*esxA*) and CFP-10 (*esxB*), become downregulated during infection in ATs ($A_i$ vs A and $E_i$ vs E, Fig. 5A). This includes the ESX-1-associated gene *espI* (*Rv3876*) that mediates ESAT-6 secretion (28), which was downregulated in both comparisons but was only significant in $A_i$ vs A (Fig.

**TABLE 1** *M.tb* genes encoding phospholipases, phosphatases, and proteases after exposure to A-ALF or E-ALF before infection[a]

| Locus tag | *M.tb* gene | A vs U | E vs UE | Product/function |
|---|---|---|---|---|
| **Phospholipases** | | | | |
| Rv2349C | **plcC** | 0.4615 | **1.4455** | Probable phospholipase C 3 PlcC/hydrolyzes sphingomyelin in addition to phosphatidylcholine |
| Rv2350C | **plcB** | −0.0507 | **1.1782** | Membrane-associated phospholipase C 2 PlcB/hydrolyzes sphingomyelin in addition to phosphatidylcholine |
| Rv2351C | **plcA** | 0.6309 | **2.1830** | Membrane-associated phospholipase C 1 PlcA (MTP40 antigen)/hydrolyzes sphingomyelin in addition to phosphatidylcholine |
| **Phosphatases** | | | | |
| Rv2234 | ptpA | 0.0922 | 0.1611 | Phosphotyrosine protein phosphatase PtpA/involved in signal transduction (via dephosphorylation) |
| Rv0153C | ptpB | −0.2016 | −0.1534 | Phosphotyrosine protein phosphatase PTPB/involved in signal transduction (via dephosphorylation) |
| Rv3310 | sapM | 0.0415 | 0.1479 | Acid phosphatase (acid phosphomonoesterase, phosphomonoesterase)/involved in cellular metabolism: acting on ester bonds |
| Rv1205 | Rv1205 | −0.2451 | −0.6111 | Conserved hypothetical protein/cytokinin riboside 5'-monophosphate phosphoribohydrolase |
| Rv0496 | Rv0496 | −0.0315 | −0.0983 | Conserved hypothetical protein/highly similar to exopolyphosphatases |
| Rv0888 | spmT | −0.3752 | 0.0113 | Sphingomyelinase/catalyzes the cleavage of sphingomyelin |
| Rv1099C | glpX | 0.0749 | −0.2112 | Fructose 1,6-bisphosphatase GlpX/involved in gluconeogenesis |
| Rv1160 | mutT2 | −0.0143 | 0.1269 | Probable mutator protein MutT2 (7,8-dihydro-8-oxoguanine-triphosphatase)/remove oxidatively damaged form of guanine |
| Rv1604 | impA | −0.0988 | −0.2920 | Probable inositol-monophosphatase ImpA (imp)/involved in inositol phosphate metabolism |
| Rv2483C | plsC | −0.4743 | −0.4488 | Probable putative L-3-phosphoserine phosphatase/involved in phospholipid biosynthesis |
| Rv2577 | **Rv2577** | 0.5807 | **1.2207** | Conserved protein/highly similar to eukaryotic phosphatases |
| Rv2701C | suhB | 0.4790 | 0.6037 | Inositol-1-monophosphatase SuhB/involved in inositol phosphate metabolism |
| Rv3137 | **Rv3137** | **1.1064** | 0.6896 | Probable monophosphatase (histidinol-phosphatase)/involved in cellular metabolism |
| **Proteases** | | | | |
| Rv3883C | mycP1 | 0.5077 | 0.2057 | Membrane-anchored mycosin MycP1 (serine protease)/proteolytic activity. Cleaves ESPB|Rv3881c |
| Rv0983 | pepD | −0.3763 | −0.5981 | Probable serine protease PepD (serine proteinase)/function unknown; possibly hydrolyzes peptides and/or proteins |
| Rv3671c | Rv3671c | 0.1355 | 0.1048 | Membrane-associated serine protease/function unknown; hydrolyses of peptides and/or proteins |
| Rv2745C | clgR | 0.0163 | −0.2812 | Transcriptional regulatory protein ClgR/controls protease systems and chaperones |
| Rv0198C | zmp1 | 0.1575 | −0.1618 | Probable zinc metalloprotease Zmp1/function unknown; hydrolyzes peptides and/or proteins |
| Rv2869C | rip | −0.4362 | −0.4997 | Membrane-bound metalloprotease/regulates intramembrane proteolysis and controls membrane composition |
| Rv1977 | Rv1977 | −0.3836 | −0.3170 | Conserved hypothetical protein/contains PS00142 neutral zinc metallopeptidases, zinc-binding region signature |
| Rv0384C | clpB | −0.3485 | −0.9697 | Probable endopeptidase ATP binding protein ClpB/thought to be an ATPase subunit of an intracellular ATP-dependent protease |
| Rv2457C | clpX | −0.3066 | −0.6417 | Probable ATP-dependent CLP protease ATP-binding subunit ClpX/it directs the protease to specific substrates |
| Rv2460C | clpP2 | −0.3139 | −0.7480 | Probable ATP-dependent CLP protease proteolytic subunit 2 ClpP2/CLP cleaves peptides that require ATP hydrolysis |
| Rv2461C | clpP1 | −0.3887 | −0.7415 | Probable ATP-dependent CLP protease proteolytic subunit 1 ClpP1/CLP cleaves peptides that require ATP hydrolysis |
| Rv2467 | pepN | 0.4829 | 0.1431 | Probable aminopeptidase N PepN (Lysyl aminopeptidase)/aminopeptidase with broad substrate specificity to several peptides |
| Rv2782C | pepR | 0.1990 | −0.0107 | Probable zinc protease PepR/function unknown; possibly hydrolyzes peptides and/or proteins |
| Rv3596C | clpC1 | 0.1509 | −0.1873 | Probable ATP-dependent protease ATP-binding subunit ClpC1/hydrolyses proteins in the presence of ATP |
| Rv2667 | clpC2 | 0.2016 | 0.4033 | Possible ATP-dependent protease ATP-binding subunit ClpC2/function unknown; possibly hydrolyzes peptides |
| Rv0983 | pepD | −0.3763 | −0.5981 | Probable serine protease PepD (serine proteinase)/function unknown; possibly hydrolyzes peptides and/or proteins |
| Rv1983 | **PE_PGRS35** | 0.3663 | **1.0598** | PE-PGRS family protein PE_PGRS35/function unknown |

*(Continued on next page)*

**TABLE 1** *M.tb* genes encoding phospholipases, phosphatases, and proteases after exposure to A-ALF or E-ALF before infection[a] (*Continued*)

| Locus tag | *M.tb* gene | A vs U | E vs UE | Product/function |
|---|---|---|---|---|
| Rv3090 | **Rv3090** | 0.9742 | **1.7766** | Conserved hypotheticals alanine and valine-rich protein/function unknown; possibly proteolytic activity |

[a]$Log_2$ FC values in ALF-exposed *M.tb* (A, Adult ALF; E, Elderly ALF) vs unexposed bacteria (UE) are shown; significant DEGs ($Log_2$ FC equal or greater than an absolute value of 1 and FDR < 0.1) are highlighted **in bold**. Gene product and function are shown as described in Mycobrowser.

5A). Conversely, *ifnT/TNT* was the only ESX-1-associated gene that was upregulated in both comparisons. The only ESX-2-related gene that was differentially expressed and downregulated in $A_i$ vs A was *espG2*, with unknown function (Fig. 5B). Furthermore, most of the ESX-3 genes (e.g., *PE5*, *esxH*, *espG3*, *mycP3*, and *eccE3*) were downregulated in both comparisons, except for *esxG* (*Rv0287*), involved in iron regulation, which was significantly downregulated only in $E_i$ vs E, as well as ESX-4-associated genes *Rv3446c* and *eccC4* (Fig. 5B). Interestingly, related to the immunomodulatory ESX-5 secretion system, two CFP-10 homologs playing crucial roles in regulating cytokine/chemokine signaling (31, 32), *esxJ* (*Rv1038C*) and *esxW* (*Rv3620c*), were found significantly upregulated in $E_i$ vs E (*esxJ*) and downregulated in $A_i$ vs A (*esxW*) (Fig. 5C). In addition, *PPE26* and *PPE65*, involved in pro-inflammatory responses (32, 37), were significantly downregulated in $E_i$ vs E, but upregulated in $A_i$ vs A (Fig. 5C). Other ESX-5-associated genes (*PPE27*, *esxI*, *Rv1042c*, *Rv1048c*, and *esxV*) were upregulated in both $A_i$ vs A, and $E_i$ vs E (Fig. 5C).

## Effects of A- vs E-ALF exposure on the expression of PDIM-related genes during *M.tb* replication in ATs

Most *ppsA-E* genes involved in producing phthiocerol and phenolphthiocerol (38) were significantly downregulated in *M.tb* exposed to A-ALF during AT infection (Table 2), except for *ppsC* which was also significantly downregulated in E-ALF-exposed bacteria. Although not significant, the transcriptional repressor, *Rv3167c*, which controls the lipid PDIM operon (20), was downregulated in E-ALF-exposed bacteria during AT infection ($E_i$ vs E, Table 2). Furthermore, in both comparisons ($E_i$ vs E and $A_i$ vs A) most of the polyketide synthases-related genes (*pks1*, *pks10*, *pks12*, *pks5,* and *pks7*), involved in PDIM biosynthesis (39), were downregulated (Table 2), as well as MmpL7 (40), involved in PDIMs and phenolic glycolipid transport across the membrane. In contrast, other PDIMs and antibiotic-related transporters were found upregulated [e.g., *drrB* and *drrC* (41, 42)] (Table 2).

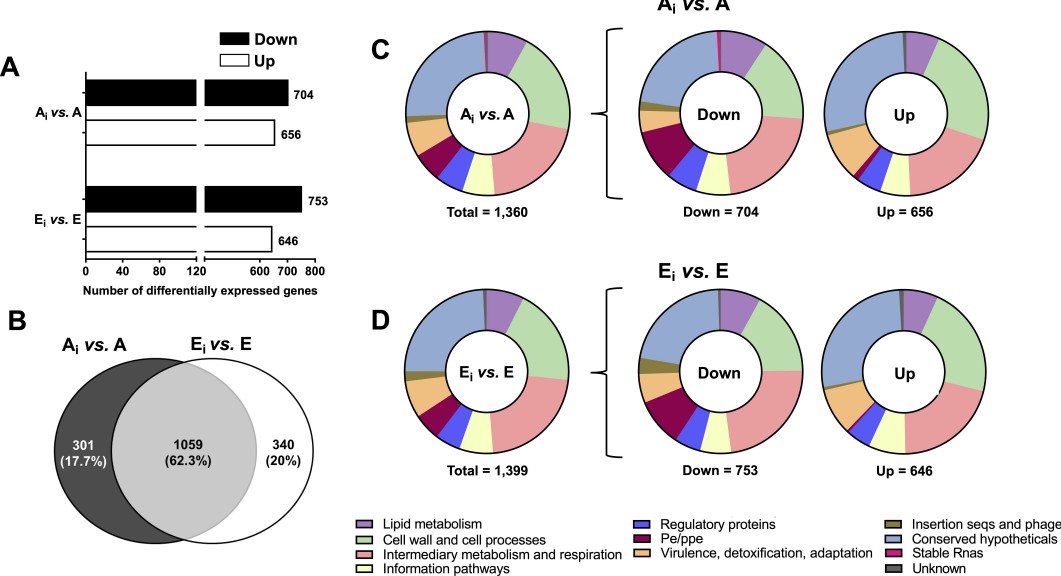

**FIG 4** Differential expression analysis of ALF-exposed *M.tb* during AT infection. (A) Number of differentially expressed genes during AT infection (ALF-exposed *M.tb* in ATs vs ALF-exposed *M.tb* before infection) obtained with the DESeq2 method in the iDEP software, with the following settings: Log$_2$ FC equal or greater than an absolute value of 1, and FDR < 0.1. Upregulated genes are shown in white bars and downregulated genes in black bars. (B) Venn diagram of DEGs during AT infection (ALF-exposed *M.tb* in ATs vs ALF-exposed *M.tb* before infection) showing unique and shared DEGs in ALF-exposed *M.tb* during AT infection. (C) Total number of DEGs distributed by functional categories during AT infection for A-ALF-exposed *M.tb* in ATs vs A-ALF-exposed *M.tb* prior infection ($A_i$ vs A) and (D) for E-ALF-exposed *M.tb* in ATs vs E-ALF-exposed *M.tb* prior infection ($E_i$ vs E) during ATs infection (represented as a percentage). Graphs were generated in GraphPad Prism v9.1.1. A, A-ALF-exposed *M.tb*; E, E-ALF-exposed *M.tb*; $A_i$, A-ALF-exposed *M.tb* during AT infection; $E_i$, E-ALF-exposed *M.tb* during AT infection.

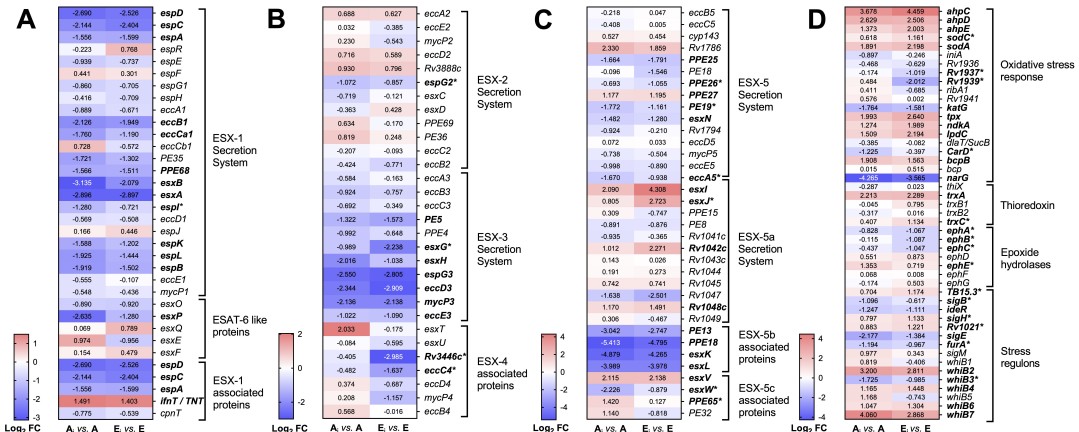

**FIG 5** *M.tb* genes associated with ESX-secretion systems and ROS defense mechanisms at 72 h post-infection (hpi) in ATs. (A) ESX-1 and ESAT-6-like proteins; (B) ESX-2, ESX-3, and ESX-4 secretion systems; (C) ESX-5 secretion system, and (D) ROS defense mechanisms, after infection of ATs with A-ALF or E-ALF-exposed *M.tb* for 72 h. Heatmap cells depict Log$_2$ FC values in ALF-exposed *M.tb* in ATs (A$_i$, Adult ALF-exposed *M.tb* during infection; E$_i$, Elderly ALF-exposed during infection) vs ALF-exposed *M.tb* before infection, upregulated: red, downregulated: blue. Genes in bold indicate significant DEGs (Log$_2$ FC equal greater than an absolute value of 1, and FDR < 0.1) for both conditions. Genes with an asterisk indicate significance in only one of the comparisons, highlighting differences between conditions. Notice the different scales used in A, B, C, and D for better visualization of the results. Heatmaps were generated in GraphPad Prism v9.1.1.

## E-ALF-exposed *M.tb* has upregulation of genes associated with the Sec (secretion) system during AT infection

Protein transport through the *M.tb* cell membrane is mainly mediated by the general secretion (Sec) pathway and the twin-arginine translocation (Tat) system, in addition to the ESX-secretion system (43–45). We assessed DEGs linked to these two *M.tb* export systems in both comparisons (E$_i$ vs E and A$_i$ vs A) (Table 3). Interestingly, E$_i$ vs E showed upregulation of *secA1*, *secE1*, and *yajC* genes during infection, the latter encoding an accessory Sec protein that aids in protein export (43) (Table 3). Furthermore, *M.tb* uses the signal recognition particle system (SRP pathway) to secrete co-translational proteins from ribosomes to the bacterial extracellular surface (46). The SRP pathway comprises three major components, FtsY, FfH, and 4.5S rRNA (46, 47). In this regard, E$_i$ vs E had upregulation of the *ffh* gene in ATs (Table 3). Lastly, we found upregulation of the *tatA* gene only in the A$_i$ vs A comparison (Table 3), which encodes for one of the three integral membrane proteins (TatA-C) of the twin-arginine translocation system, important to the export of folded proteins across the bacterial cell envelope (44).

## Upregulation of E-ALF-exposed *M.tb* genes encoding oxidative stress defense mechanisms during AT infection

*M.tb* has developed strategies that mitigate oxidative stress and overcome the microbicidal capacity within the host (48, 49). We next explored DEGs linked to reactive oxygen species (ROS) defense mechanisms (Fig. 5D; Fig. S4). While *ahpC*, *ahpD*, *ahpE*, *lpdC*, *trxA*, and *sodA* genes were upregulated in both comparisons (E$_i$ vs E and A$_i$ vs A), *sodC* and *trxC* were significantly upregulated only in the E$_i$ vs E (Fig. 5D), all associated with response to oxidative stress and defense against the microbicidal capacity of the host (50).

In addition, epoxide hydrolases, sigma factors, and WhiB proteins are all characterized as additional *M.tb* detoxifying strategies to withstand adverse conditions (51–53). Our findings indicate that during infection E-ALF-exposed *M.tb* downregulates some epoxide hydrolases genes (*ephA*, *ephB,* and *ephC*) (Fig. 5D). Conversely, sigma factor *sigH*, which is considered a crucial *M.tb* transcriptional factor responding to oxidative, nitrosative, and heat stressors (52, 54), was found upregulated in E$_i$ vs E (Fig. 5D), while sigma factor B (*sigB*), which plays comparable roles (55), was downregulated in A$_i$ vs A (Fig. 5D). Finally, the WhiB family are redox-sensing transcription factors in which the three members (WhiB3, WhiB4, and WhiB7) are responsible in maintaining redox homeostasis

**TABLE 2** *M.tb* genes related to lipid PDIMs biosynthesis and translocation during infection in ATs[a]

| Locus tag | *M.tb* gene | A_i vs A | E_i vs E | Product/function |
|---|---|---|---|---|
| Rv3546 | fadA5 | 0.1227 | 0.0890 | Probable acetyl-CoA acetyltransferase FadA5/involved in lipid degradation |
| Rv2930 | fadD26 | −0.3769 | −0.0480 | Fatty-acid-AMP ligase FadD26/fatty-acid-AMP synthetase |
| Rv2931 | **ppsA** | **−1.1493** | −0.7337 | Phenolphthiocerol synthesis type-I polyketide synthase PpsA/phthiocerol dimycocerosate (dim) biosynthesis |
| Rv2932 | **ppsB** | **−1.4757** | −0.8243 | Phenolphthiocerol synthesis type-I polyketide synthase PpsB/phthiocerol dimycocerosate (dim) biosynthesis |
| Rv2933 | **ppsC** | **−2.0269** | **−1.3387** | Phenolphthiocerol synthesis type-I polyketide synthase PpsC/phthiocerol dimycocerosate (dim) biosynthesis |
| Rv2934 | **ppsD** | **−1.6274** | −0.9475 | Phenolphthiocerol synthesis type-I polyketide synthase PpsD/phthiocerol dimycocerosate (dim) biosynthesis |
| Rv2935 | **ppsE** | **−1.3829** | −0.7921 | Phenolphthiocerol synthesis type-I polyketide synthase PpsE/phthiocerol dimycocerosate (dim) biosynthesis |
| Rv2936 | drrA | −0.7882 | −0.0797 | Daunorubicin-dim-transport ATP-binding protein ABC transporter DrrA/transport of phthiocerol dimycocerosate |
| Rv2937 | **drrB** | **1.1364** | 1.5274 | Daunorubicin-dim-transport integral membrane protein ABC transporter DrrB/transport of phthiocerol dimycocerosate |
| Rv2938 | **drrC** | **1.2649** | 1.3778 | Probable daunorubicin-dim-transport integral membrane protein integral membrane protein ABC transporter DrrC/transport of phthiocerol dimycocerosate |
| Rv2939 | papA5 | −0.0384 | 0.4154 | Possible conserved polyketide synthase-associated protein PapA5/phthiocerol dimycocerosate (dim) biosynthesis |
| Rv2940C | mas | −0.3390 | 0.0106 | Probable multifunctional mycocerosic acid synthase membrane-associated Mas |
| Rv2941 | fadD28 | 0.0311 | 0.7592 | Fatty-acid-AMP ligase FadD28 (fatty-acid-AMP synthetase)/phthiocerol dimycocerosate (dim) biosynthesis |
| Rv2942 | **mmpL7** | **−1.4642** | −1.2812 | Conserved transmembrane transport protein MmpL7/transport of phthiocerol dimycocerosate |
| Rv3167C | Rv3167c | 0.0062 | −0.5682 | Probable transcriptional regulatory protein/probably TetR-family |
| Rv3169 | Rv3169 | −0.2734 | 0.2865 | Conserved protein/function unknown |
| Rv2946C | **pks1** | **−2.1670** | −1.7035 | Probable polyketide synthase Pks1/polyketide synthase possibly involved in lipid synthesis |
| Rv2947C | **pks15** | **−1.9372** | −0.9599 | Probable polyketide synthase Pks15/polyketide synthase possibly involved in lipid synthesis (biosynthesis of phthiocerol) |
| Rv1660 | **pks10** | **−2.3674** | −2.4025 | Chalcone synthase Pks10/biosynthesis of secondary metabolites |
| Rv2048C | **pks12** | **−1.4493** | −1.2676 | Polyketide synthase Pks12/biosynthesis of mannosyl-beta-1-phosphomycoketide |
| Rv1527C | **pks5** | **−1.3134** | −1.5411 | Probable polyketide synthase Pks5/involved in polyketide metabolism |
| Rv1661 | **pks7** | **−4.1049** | −3.6883 | Probable polyketide synthase Pks7/involved in polyketide metabolism |

[a]Log₂ FC values in ALF-exposed *M.tb* in ATs (A_i, Adult ALF-exposed *M.tb* during infection; E_i, Elderly ALF-exposed *M.tb* during infection) vs ALF-exposed *M.tb* before infection are shown; significant DEGs (Log₂ FC equal or greater than an absolute value of 1 and FDR < 0.1) are highlighted in bold. Gene product and function are shown as described in Mycobrowser.

**TABLE 3** *M.tb* genes associated with the Sec secretion and Tat translocation systems during replication in ATs[a]

| Locus tag | *M.tb* gene | $A_i$ vs A | $E_i$ vs E | Product/function |
|---|---|---|---|---|
| **Sec secretion system** | | | | |
| Rv3240C | **secA1** | 0.5768 | **1.2521** | Probable preprotein translocase SecA1 1 subunit/involved in protein export |
| Rv1821 | secA2 | −0.4800 | 0.1331 | Possible preprotein translocase ATPase SecA2/involved in protein export |
| Rv2587C | secD | −0.1348 | 0.0324 | Probable protein-export membrane protein SecD |
| Rv0638 | **secE1** | 0.6131 | **1.1954** | Probable preprotein translocase SecE1/essential for protein export |
| Rv2586C | secF | −0.5924 | −0.3857 | Probable protein-export membrane protein SecF/involved in protein export |
| Rv1440 | **secG** | **1.4243** | **1.9535** | Probable protein-export membrane protein (translocase subunit) SecG/participates in an early event of protein translocation |
| Rv0732 | **secY** | **2.0271** | **2.6184** | Probable preprotein translocase SecY/essential for protein export |
| Rv2916C | **ffh** | 0.5952 | **1.0993** | Probable signal recognition particle protein Ffh/necessary for efficient export of extra-cytoplasmic proteins |
| Rv2921C | ftsY | −0.1439 | 0.3242 | Probable cell division protein FtsY (SRP receptor)/reception and insertion of a subset of proteins at the membrane |
| Rv2588C | **yajC** | 0.7343 | **1.3427** | Probable conserved membrane protein secretion factor YajC/involved in secretion apparatus |
| Rv3921C | yidC | 0.6739 | 0.7670 | Probable conserved transmembrane protein/cell wall and cell processes |
| **Tat secretion system** | | | | |
| Rv2094C | **tatA** | **1.4052** | 0.9527 | Sec-independent protein translocase membrane-bound protein TatA/involved in protein export |
| Rv1224 | tatB | 0.6343 | 0.6229 | Probable protein TatB/twin-arginine translocation (Tat) system |
| Rv2093C | tatC | 0.5748 | 0.0482 | Sec-independent protein translocase transmembrane protein TatC/involved in protein export |

[a]Log$_2$ FC values in ALF-exposed *M.tb* in ATs vs ALF-exposed *M.tb* before infection are shown; significant DEGs (Log$_2$ FC equal or greater than an absolute value of 1 and FDR < 0.1) are highlighted in bold. Gene product and function are shown as described in Mycobrowser.

(53). Our results indicate that *whiB2, whiB4, whiB6,* and *whiB7* were upregulated in both comparisons, whereas *whiB3* was downregulated only in $A_i$ vs A (Fig. 5D). Other stress response-related genes were uniquely upregulated ($E_i$ vs E: *TB15*.3 and *Rv1021*; $A_i$ vs A: *ephE*) or downregulated ($E_i$ vs E: *Rv1937* and *Rv1939*; $A_i$ vs A: *furA* and *CarD*) (Fig. 5D), while others were shared between both comparisons (upregulated: *tpx* and *bcpB*; downregulated: *katG*, *narG*, *ideR*, and *sigE*) (Fig. 5D).

## Upregulation of other virulent factors in E-ALF-exposed *M.tb* during AT infection

Additional virulence determinants that may contribute to *M.tb* pathogenesis including proteases (*zmp1*, *clpP1,* and *pepN*) (56, 57), lipases (*lipE*) (58), iron efflux pumps (*MmpL4*) (59) and adhesins (*apa*, *fbpB*, *glnA1*, *gap,* and *pstS1*) (60) were found upregulated in $E_i$ vs E, as well as other genes associated with mycobacterial survival (*glpX* and *Rv1096*) (61, 62) (Fig. S4). Other upregulated or downregulated DEGs in $E_i$ vs E or $A_i$ vs A or both comparisons are depicted in Fig. S4.

## DISCUSSION

The elderly population is considered a large reservoir for *M.tb* infection due to their increased vulnerability to active TB disease or reactivation of latent TB (6, 8). We have published that the lung mucosa or ALF from elderly individuals (65+ years old) is highly oxidized and constitutes a pro-inflammatory environment with dysfunctional innate proteins (12–14) and that *M.tb* replicates faster upon contacting the elderly lung mucosa (E-ALF) in human macrophages and can replicate to a higher bacterial load in mice, leading to more extensive lung tissue damage (13). More recently, we reported that *M.tb* exposure to E-ALF enhances intracellular growth in ATs and alters endosomal trafficking with increased bacterial translocation into the cytosol (15). Our findings herein suggest that age-associated ALF changes lead to an increased predisposition of the elderly population to *M.tb* infection. Thus, it is critical to understand how exposure to human ALF from adult or elderly individuals changes the *M.tb* metabolic status. In this study, we

investigated the transcriptional profile of *M.tb* when exposed to human ALF (A-ALF or E-ALF) before and during intracellular replication in ATs.

Our DEG analysis prior to infection shows about three times more DEGs in E-ALF-exposed *M.tb* compared to A-ALF-exposed *M.tb* (Fig. 2A), which could be explained by the different ALF composition in elderly individuals, resulting in different *M.tb* recognition by the host cells and infection outcomes. However, once ALF-exposed *M.tb* is intracellular in ATs, *M.tb* bacilli are exposed to the same microenvironment and seem to start normalizing their gene expression, which would explain why the number of DEGs in E-ALF vs A-ALF-exposed *M.tb* during infection is similar (although not identical) (Fig. 4A). Although the DEG numbers observed are similar, ~40% of the DEGs are specific to either A-ALF or E-ALF exposure during AT infection, indicating that transcriptional responses during infection are qualitatively different, as seen by differences in *M.tb* growth and translocation into the cytosol in the case of E-ALF-exposed *M.tb* (15), among others.

Our DEG analysis specifically revealed upregulation of genes associated with the ESX-secretion systems and phospholipases in E-ALF-exposed *M.tb* compared to unexposed *M.tb* before infection, but not in A-ALF-exposed *M.tb* (Fig. 6). During AT infection, E-ALF-exposed *M.tb* also showed altered expression of virulence-associated genes related to the ESX-secretion systems, as well as several genes associated with PDIMs metabolism and ROS defense strategies, among others (Fig. 6). Our findings suggest that *M.tb* takes advantage of E-ALF exposure by driving metabolic changes to avoid the intrinsic antimicrobial mechanisms of the host with potential implications in infection outcome.

Indeed, *M.tb* has evolved a vast range of virulence factors implicated in lipid metabolism and protein transport, signal transduction, nutrient uptake, immune evasion, and resistance to microbicidal activity, among others (48, 63). For instance, *M.tb* employs five ESX protein secretion systems (ESX-1 to ESX-5). In general, full *M.tb* virulence requires the presence of ESX-1, ESX-3, and ESX-5 (25). However, the ESX-1 secretion system is considered one of the key virulence factors due to the secretion of the 6 kDa Early Secreted Antigenic Target (ESAT-6/EsxA) and its partner protein, the 10 kDa Culture Filtrate Protein (CFP-10/EsxB), associated with phagosomal rupture and bacterial translocation into the host cytosol (22, 25, 64).

In this regard, for *M.tb* exposure to ALF prior to infection, our results indicate that although the genes encoding ESAT-6 (*esxA*) and CFP-10 (*esxB*) are not significantly upregulated after exposure to either A- or E-ALF, *esxE* and *cpnT* are slightly more upregulated in the E-ALF group. Both genes are encoded by the *cpnT* operon, which may allow TB necrotizing toxin (TNT) secretion (CpnT/TNT) to contribute to the permeabilization of the phagosomal membrane (29, 30). Particularly, the EsxF/EsxE heterodimer generates persistent membrane-spanning channels that exhibit pore-forming activity

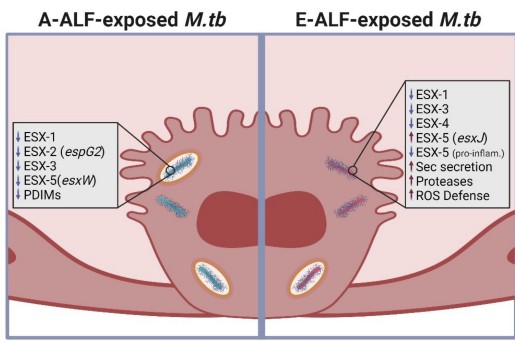

**FIG 6** Schematic overview of the main findings in this study. Transcriptional profile of ALF-exposed-*M.tb* prior to and during intracellular replication in ATs. Figure created using BioRender (https://biorender.com/).

(65). This highlights how the ESX family in *M.tb* has paralogous gene pairs, in which proteins encoded frequently form a 1:1 heterodimer (e.g., EsxA-EsxB) (66) and cause lysis of the phagosome and bacterial translocation into the cytosol (21, 22). However, *M. abscessus* lacks the ESX-1 secretion system, but instead, has the ESX-4 secretion system with a similar role (27, 31). Interestingly, E-ALF-exposed *M.tb* (prior to infection) shows upregulation of genes associated with the ESX-4 secretion system (*Rv3446c*, *eccC4*, *eccD4*, and *mycP4*). Other ESX-5-related genes are differentially expressed, such as *PPE18* (upregulated in A-ALF-exposed *M.tb*) and *PPE65* (upregulated in E-ALF-exposed *M.tb*) involved in anti-inflammatory and pro-inflammatory signaling, respectively (32).

In addition, all phospholipase-related genes (*plcC*, *plcB*, and *plcA*), which mediate hydrolytic reactions with cytotoxic effects on macrophages (67), were upregulated when *M.tb* was exposed to the E-ALF but not A-ALF (prior to infection). Interestingly, the phosphatidylinositol-specific phospholipase C (PI-PLC/*plcA*) and the broad-range phospholipase C (PC-PLC/*plcB*) enzymes, together with the pore-forming toxin listeriolysin O, are used by the intracellular pathogen *Listeria* to break the phagosomal membrane (33, 34). In *M.tb*, these virulence phospholipase homologs confer higher resistance to alveolar macrophage microbicidal activity (67) and are suggested to participate in the breach of vacuolar membranes (68). Additionally, some virulent-related phosphatases (e.g., *Rv2577*) and protease genes (e.g., *PE_PGRS35* and *Rv3090*) were also upregulated in the E vs UE comparison. *Rv2577* is characterized as an alkaline phosphatase/phosphodiesterase enzyme, which plays a role in *M.tb* pathogenicity by promoting increased replication and phagosomal arrest in early endosomes (36), and *Rv3090* induces late cell apoptosis, resulting in bacterial dissemination and organ damage in mice (35). Overall, these results highlight how exposure to the human elderly lung mucosa prior to infection drives the upregulation of virulence-related genes in *M.tb*, potentially priming the bacterium to upregulate specific functions involved in mitigating the host immune response and ready its machinery to permeabilize phagosomes before contacting host cells.

We next investigated the effects of E-ALF exposure on *M.tb* metabolism during infection of ATs. In contrast with previous studies that reported upregulation of ESX-1-associated genes (69), our results indicate that most ESX-1 genes are downregulated after exposure to ALF (either A- or E-ALF) during AT infection. Even though the expression of *esxA* is not upregulated, its levels could be influenced by additional ESX proteins that mediate its secretion. For instance, A-ALF-exposed *M.tb* had downregulation of *EspI*, which regulates ESAT-6 secretion, impacting its availability and potential levels of phagosomal disruption.

However, our findings revealed the upregulation of some DEGs associated with ESX-5 during AT infection. Interestingly, *esxJ*, a homolog of CFP-10/*esxB*, was significantly upregulated in E-ALF-exposed *M.tb* during infection, while the other homolog *esxW* was significantly downregulated in A-ALF- but not in E-ALF-exposed *M.tb*. Further studies are needed to verify whether these homologs, mainly *esxJ*, may play a similar role to *esxB* from the ESX-1 system, leading to bacterial escape into the cytosol. ESX-5 immunomodulatory proteins, PPE26 and PPE65, responsible for pro-inflammatory responses in macrophages (32, 37), were both downregulated in E-ALF-exposed *M.tb* during replication in ATs, suggesting potential mechanisms of *M.tb* to avoid pro-inflammatory responses within the host leading to increased intracellular survival, consistent with our prior findings (15).

One of the main *M.tb* virulence factors is PDIMs, which play a broad range of roles during infection (70–72). Recent evidence suggests that PDIMs cooperate with *esxA* creating pores in the phagosomal membrane to allow access of *M.tb* and secreted proteins into the host cell cytosol, ultimately leading to cell apoptosis (20, 63, 71). In our data set, A-ALF-exposed *M.tb* during infection in ATs had significant downregulation of genes encoding the phenolphthiocerol synthesis type-I polyketide synthase (*ppsA*, *ppsB*, *ppsC*, *ppsD*, and *ppsE*) that mediate PDIMs biosynthesis, while E-ALF-exposed *M.tb* only showed significant downregulation of *ppsC*. Polyketide synthases *pks1*, *pks10*, *pks12*,

*pks5*, and *pks7*, involved in the production of PDIMs, were downregulated in both A- and E-ALF-exposed *M.tb* during infection. Conversely, PDIM transport genes *drrB* and *drrC* were the only genes significantly upregulated in both groups. Finally, the transcriptional repressor *Rv3167c*, which regulates the PDIM operon, was marginally downregulated in E-ALF-exposed bacteria. Interestingly, the loss-of-function of *Rv3167c* correlates with the capacity of *M.tb* to escape the phagosome (20).

In addition, mycobacteria use a wide range of secretion systems including the general secretion (Sec) pathway (SecA1 secretory pathway and the alternative SecA2-operated pathway) and a twin-arginine translocation system (Tat system) (43–45, 73). Our results indicate that *secA1*, essential for *M.tb* growth and responsible for exporting most proteins through the SecYEG complex (43, 45), and *secE1* genes are significantly upregulated during infection of ATs with E-ALF-exposed *M.tb* but not in A-ALF-exposed *M.tb*, whereas *secG* and *secY* genes are upregulated regardless of the ALF exposure condition. The *yajC* gene, which encodes an auxiliary Sec protein (43), was also upregulated only in E-ALF-exposed *M.tb*. Regarding the Tat system, we observed upregulation of the *tatA* gene during infection of ATs by A-ALF-exposed bacteria, which encodes for one of its integral membrane proteins (TatA-C) (44). Overall, these results suggest an increase in protein transport in intracellular *M.tb* that has been previously exposed to E-ALF when compared to A-ALF-exposed *M.tb*.

Furthermore, mycobacteria have natural mechanisms to combat the microbicidal capacity of host cells, such as the production of virulence factors against oxidative and nitrosative stresses (48, 49). Indeed, *M.tb* genes encoding catalase-peroxidase (*katG*), superoxide dismutase (*sodA*), alkyl peroxide reductase (*ahpCDE*), thioredoxin (*thiX* and *trxB1C*), or thioredoxin reductase (*trxB2*) are upregulated in response to nitrosative and oxidative stressors (50). Several genes that encode ROS defense mechanisms were upregulated in both A-ALF and E-ALF-exposed *M.tb* during AT infection (e.g., *ahpC*, *ahpD*, *ahpE*, *trxA*, *lpdC*, and *sodA*). However, *sodC* and *trxC* were upregulated only in E-ALF-exposed *M.tb*. Superoxide dismutase (SOD) is a radical scavenger that reduces NO and oxygen radicals' production, allowing *M.tb* to tolerate oxidative stress and survival within host cells (74), whereas thioredoxin (Trx) is involved in the maintenance of cellular redox equilibrium, contributing to *M.tb* resistance to phagocytosis (75). Additionally, sigma factors *sigH* and *sigB*, essential for bacterial adaptation to oxidative, nitric oxide, and heat stress (76, 77), were significantly upregulated in E-ALF-exposed *M.tb* but downregulated in A-ALF-exposed *M.tb*. Furthermore, WhiB proteins perform a variety of functions including regulating redox homeostasis. Although *whiB2*, *whiB4*, *whiB6,* and *whiB7* were upregulated in both A-ALF and E-ALF-exposed *M.tb* during infection of ATs, *whiB3*, necessary to mitigate redox stress caused by ROS and reactive nitrogen species (RNS) and to survive within macrophages (53), was downregulated solely in the A-ALF condition.

Other virulence factors are required for *M.tb* growth and pathogenesis, such as proteases, lipases, and adhesins. The proteases *zmp1* and *clpP1,* and the lipase *lipE,* were upregulated in E-ALF-exposed *M.tb* during AT infection. Metalloprotease Zmp-1 is involved in inflammasome inactivation and phagosome maturation arrest and is essential for intracellular survival (78, 79), and clpP1 is critical for *M.tb* growth (80). Similarly, LipE with lipase/esterase activity is induced under stress conditions and is essential for *M.tb* intracellular growth *in vivo* (58). Lastly, several adhesins such as *apa*, *fbpB*, *glnA1*, *gap,* and *pstS1* were upregulated in E-ALF-exposed *M.tb* during infection. These adhesins are associated with adherence-mediating elements and strategies that permit *M.tb* attachment to the host (60).

Finally, a potential limitation of this study is the use of A- and E-ALF pools instead of individual ALFs. We chose this strategy to reduce human-to-human variability within each ALF group studied and focus on the overall effects of A- and E-ALF exposure rather than donor-specific changes. This decision was made based on our previous publication, where human E-ALFs presented similar composition (14); however, the possibility exists that if one (or more) ALFs from the pool have a very distinct composition it

could confound data and interpretations undermining the generalizability of the results obtained. In this context, we understand that human variability strongly influences the outcomes of *M.tb* infection and thus, the impact of individual A- and E-ALFs on *M.tb* adaptation to the lung environment will be investigated in further studies. We also acknowledge that the use of A549 cells did not allow us to study the variability in primary human ATs, which might also impact *M.tb*-host cell interactions and infection outcomes. However, we chose to remove this source of variability to focus on the contribution of ALF from elderly vs adult populations in *M.tb*'s transcriptional adaptation before and during infection. Finally, we note here that for one of the experimental conditions (E-ALF-exposed *M.tb* before infection or E-ALF) we only have $n = 2$ replicates, since one of the replicates did not pass the QC (RIN < 7) and was removed from the study.

By studying how *M.tb* adapts to E-ALF, we were able to provide potential mechanisms, that might work in conjunction, explaining how E-ALF-exposed *M.tb* could escape into the cytosol by altering the expression of genes associated with many different virulence mechanisms such as ESX-secretion systems, Sec secretion systems, PDIMs metabolism, and ROS defense strategies, among others. Future functional studies will be required to delineate how these mechanisms potentially contribute to the susceptibility of the elderly population to TB pathogenesis.

## MATERIALS AND METHODS

### Human subjects and ethics statement

Human subject studies were conducted following stringent guidelines established by the US Code of Federal and Local Regulations (OSU Institutional Review Board IRB numbers 2012H0135 and 2008H0119). After receiving informed written consent, bronchoalveolar lavage fluid (BALF) from healthy adults (18 to 45 years old) and elderly (≥69 years old) participants from both sexes were collected. There was no consideration given to race or ethnicity. The following comorbidities were excluded: smokers (current or less than one month), excessive alcohol users, non-injection/injection recreational drug use, active pneumonia, asthma, bilateral cancer, and on chemotherapy, chronic obstructive pulmonary disease, pre-diabetes/diabetes (hemoglobin A1c higher than 5.7% within the last three months or a fasting blood glucose level of higher than 110 mg/dL), body mass index equal or higher than 40, hepatitis, human immunodeficiency HIV/AIDS, immunosuppression or taking non-steroidal anti-inflammatory agents (TNF antagonists), leukemia/lymphoma, liver failure, renal failure, rheumatoid arthritis, pregnancy (or gave birth less than three months ago), taken antibiotics recently and/or regular treatment with over the counter anti-inflammatory medications.

### Human ALF isolation

Human BALF from healthy young and elderly donors was concentrated to obtain ALF at its physiological concentration within the human lung (1 mg/mL of phospholipid) as we reported in detail (10, 12, 13, 15, 24, 81).

### AT culture

The human ATs type II-like cell line A549 (ATCC CCL-185), a lung cancer cell line with many AT type II cell features (used as a model of ATs), was used for all cell culture experiments. Cell cultures were prepared as described (15, 24) with minor changes. Briefly, the A549 cell line was cultured at 37℃ with 5% $CO_2$ in culture media [DMEM/F12 supplemented with 10% FBS (Atlas Biologicals, Fort Collins, CO)] and 1% PenStrep (Sigma, St. Louis, MO). One week before *M.tb* infection, cells were kept in antibiotic-free growth media. Cells were infected after four passages from the ATCC stock.

## *M.tb* culture and exposure to human ALF

*M.tb* Erdman GFP strain (kindly provided by Dr. Marcus Horwitz, UCLA) was grown as described (10). Single-cell bacterial suspensions of $1 \times 10^9$ bacteria were generated in sterile endotoxin-free human isotonic 0.9% NaCl as we reported (13, 15, 24), and bacterial pellet was obtained after centrifugation (13,000 $\times$ *g*, 10 min) and exposed to a 50 µL pool of A-ALF (*n* = 4 different human donors) and E-ALF (*n* = 4 different human donors) for 12 h at 37°C with 5% $CO_2$ (Fig. S1). Following exposure, bacteria were thoroughly washed with 0.9% NaCl to remove any unbound ALF, and bacterial pellet obtained after centrifugation (13,000 $\times$ *g*, 10 min) was either stored at −80°C for further RNA extraction [after incubation in RNAProtect (Qiagen, Hilden, Germany) for 10 min at room temperature, RT] or suspended in culture media for AT infection. To assess *M.tb* viability and specifically to demonstrate that there are no differences in viable bacterial counts between E-ALF vs A-ALF-exposed *M.tb*, ALF-exposed *M.tb* inoculums were serially diluted in 7H9 broth and plated on 7H11 agar supplemented with OADC (oleic acid-albumin-dextrose-catalase) enrichment. Unexposed *M.tb* were processed in parallel and used as control (bacterial pellet incubated for 12 h at 37°C without ALF). Three independent experiments were performed (prior to and during AT infection); three different exposures to the same pools of ALF, using single bacterial suspensions of newly fresh harvested bacteria grown in 7H11 agar. Experimental conditions are shown in Fig. 1A.

## *M.tb* infection of ATs

*M.tb* infection of ATs was performed as we described in detail elsewhere (15, 24), using a multiplicity of infection (MOI) of 100:1 in $4 \times 150$ cm$^2$ flasks per condition (Fig. S1), which maintains an intact AT monolayer. After 2 h of infection, cells were washed and incubated with gentamycin (50 µg/mL)-supplemented media for 1 h to eliminate extracellular bacteria. ATs were then rinsed with culture media and cultured for up to 72 h post-infection (hpi) in culture media supplemented with 10 µg/mL of gentamicin. The use of gentamycin in the culture media kills extracellular bacteria that is being released after cells are lysed during infection, and allows us to perform our downstream transcriptome analysis focusing only on intracellular bacteria. Experimental conditions are shown in Fig. 1B. The chosen time point and MOI provide the quantity of bacterial RNA within ATs (as non-professional phagocytes) required to conduct the analysis described in this work. The stability of the infected monolayer was verified by microscopy before RNA extraction.

## *M.tb* RNA extraction, library prep, and RNA sequencing

After ALF exposure, bacterial pellets (before AT infection) were incubated in RNAProtect (Qiagen, Hilden, Germany) for 10 min at RT, centrifuged, and stored at −80°C until further processing. RNA extraction was performed using the Quick-RNA Fungal/Bacterial Miniprep kit (Zymo Research, Irvine, CA, USA), following the manufacturer's instructions and as we described (82). Briefly, bacterial pellets were suspended in RNA lysis buffer and transferred to a ZR BashingBead Lysis tube containing 0.1 mm and 0.5 mm ceramic beads. Samples underwent a bead beating procedure in a Disruptor Genie (10 cycles of 1 min at maximum speed with 1 min intervals on ice), facilitating the breaking down of the *M.tb* cell envelope. RNA in the supernatant was isolated using the Zymo-spin columns, including an in-column DNAse I treatment, and eluted in nuclease-free water. A second DNAse treatment was then performed using TURBO DNAse (Thermo Fisher Scientific, Waltham, MA, USA). RNA was further purified using the RNA Clean & Concentrator kit (Zymo Research, Irvine, CA, USA), and RNA concentration and quality were assessed using the Qubit 4 Fluorometer with the HS RNA kit (Thermo Fisher Scientific, Waltham, MA, USA). RIN numbers were acquired using a 4200 TapeStation System (Agilent, Santa Clara, CA, USA).

Infected ATs (72 hpi) were washed twice with culture media and then lysed with GTC solution (4 M guanidinium thiocyanate, 0.5% sodium N-lauryl sarcosine, 25 mM

tri-sodium citrate, 0.1 M 2-mercaptoethanol, 0.5% Tween 80, pH 7.0) (69) for 5 min at RT and centrifuged at 5,000 × *g* for 30 min at RT to recover bacteria. *M.tb* pellets were washed in GTC lysis solution to combine the content from all flasks (4 × 150 cm² flasks/condition) into one single tube, centrifuged, and suspended in RNA lysis buffer. RNA was then extracted using the Quick-RNA Fungal/Bacterial Miniprep kit, treated with TURBO DNAse, and purified with the RNA Clean & Concentrator kit, as detailed above.

A whole-stranded transcriptome RNA library was generated using the Zymo-Seq RiboFree Total RNA kit (Zymo Research, Irvine, CA, USA), following the manufacturer's protocol. RNA (250 ng) was used as input to construct the library, which included a rRNA depletion step of 30 min followed by 15 cycles of PCR. Using a NextSeq 500 Mid Output Illumina platform, a 75 bp paired-end (PE) sequencing was conducted (Fig. 1C). All samples were sequenced in the same flow cell to avoid sequencing batch effects. RNA sequencing was performed on three independent experiments ($n = 3$ per condition), except for E-ALF-exposed *M.tb* before infection ($n = 2$), where one of the replicates was not included in the study due to low RNA concentration and quality (RIN < 7).

## Data analysis

After RNA sequencing, resulting reads were trimmed and aligned to the *M.tb* $H_{37}R_v$ reference genome (Genbank NC_000962.3) (83–85) with the STAR aligner v2.5.3a in the Partek Flow Genomic Analysis software (v9.0.20.0202, Partek Inc., Chesterfield, MO, USA), using default settings. Raw read count data were analyzed in iDEP.94 (http://bioinformatics.sdstate.edu/idep94/) (86). Briefly, data were filtered to remove poorly expressed genes. DEGs that were significant between experimental conditions (Fig. 1) were identified using DESeq2 (87) with the following settings: fold change (FC) of 2 (or $Log_2$ FC equal or greater than an absolute value of 1), and FDR cut-off of 0.1. The Benjamini-Hochberg procedure was used to adjust the FDR, and fold changes were calculated using an Empirical Bayes shrinkage approach (87). Functional categories for DEGs were extracted from Mycobrowser (https://mycobrowser.epfl.ch/) (88). Venn diagrams were generated in Venny 2.1.0 (https://bioinfogp.cnb.csic.es/tools/venny/index.html) (89) or GraphPad Prism Version 9.5.1 Heatmaps were generated in GraphPad Prism Version 9.5.1.

## ACKNOWLEDGMENTS

We thank Dr. Marcus Horwitz (University of California, Los Angeles) for kindly providing the GFP-*M.tb* Erdman strain. We acknowledge Clinton Christensen at the Texas Biomed Molecular Core and the BSL-3 Operations Program at Texas Biomed for their services and support.

This study was supported by the National Institute on Aging (NIA), National Institutes of Health (NIH) (Grant number P01 AG-051428 to J.B.T., L.S.S., S.H.W., and J.T. and F99 AG-079802 to A.M.O.-F.), and partially supported by the Robert J. Kleberg, Jr. and Helen C. Kleberg Foundation (J.B.T.) and NIH award AI-136831 (L.S.S.). A.M.O.-F. was partially supported by the Douglass Graduate Fellowship at Texas Biomed. A.M.O.-F., A.A.-G., A.G.-V., L.S.S., J.T., and J.B.T. are part of the Interdisciplinary NextGen Tuberculosis Research Advancement Center (IN-TRAC) at Texas Biomed, which is supported by the NIAID/NIH under the award number P30 AI-168439. Sequencing data wasdata were generated in the Texas Biomed Molecular Core, which is supported and subsidized by institutional resources, and the Genome Sequencing Facility which is supported by UT Health San Antonio, NIH-NCI P30 CA054174 (Cancer Center at UT Health San Antonio) and NIH Shared Instrument grant S10OD030311 (S10 grant to NovaSeq 6000 System), and CPRIT Core Facility Award (RP220662). The content is solely the responsibility of the authors and does not necessarily represent the official views of the National Institutes of Health.

Study design and conceptualization, A.M.O.-F. and J.B.T.; Experimental procedures, A.M.O.-F., A.A.-G., and A.G.-V.; Acquisition of data and formal analyses, A.M.O.-F. and A.A.-G.; Collect BAL fluid, R.E.M. and S.H.W.; RNA library preparation and initial filtering/alignment of sequencing reads, J.G.; Writing manuscript, A.M.O.-F. and J.B.T.;

Visualization and BioRender figures, A.M.O.-F.; Writing—review and editing, A.M.O.-F., A.A.-G., J.B.T., J.T., Y.W., and L.S.S.; Funding acquisition, A.M.O.-F., J.T., and J.B.T. All authors provided comments/edits and approved the final version of this manuscript.

## AUTHOR AFFILIATIONS

[1]Population Health and Host-Pathogen Interactions Programs, Texas Biomedical Research Institute, San Antonio, Texas, USA

[2]Integrated Biomedical Sciences Program, University of Texas Health Science Center at San Antonio, San Antonio, Texas, USA

[3]International Center for the Advancement of Research & Education (I • CARE), Texas Biomedical Research Institute, San Antonio, Texas, USA

[4]Department of Internal Medicine, Infectious Disease Division, The Ohio State University, Columbus, Ohio, USA

[5]Department of Surgery, The Ohio State University, Columbus, Ohio, USA

[6]Department of Molecular Microbiology and Immunology, South Texas Center for Emerging Infectious Diseases, University of Texas at San Antonio, San Antonio, Texas, USA

## PRESENT ADDRESS

Joanne Turner, Abigail Wexner Research Institute at Nationwide Children's Hospital, Columbus, Ohio, USA

## AUTHOR ORCIDs

Angélica M. Olmo-Fontánez  http://orcid.org/0000-0002-7255-9714
Anna Allué-Guardia  http://orcid.org/0000-0002-1941-2545
Andreu Garcia-Vilanova  http://orcid.org/0000-0002-2385-8718
Jordi B. Torrelles  http://orcid.org/0000-0001-7702-5941

## AUTHOR CONTRIBUTIONS

Angélica M. Olmo-Fontánez, Conceptualization, Data curation, Formal analysis, Funding acquisition, Investigation, Methodology, Project administration, Validation, Visualization, Writing – original draft, Writing – review and editing | Anna Allué-Guardia, Data curation, Formal analysis, Investigation, Methodology, Validation, Writing – original draft, Writing – review and editing | Andreu Garcia-Vilanova, Investigation | Jeremy Glenn, Investigation | Shu-Hua Wang, Funding acquisition, Resources, Writing – review and editing | Robert E. Merritt, Resources | Larry S. Schlesinger, Funding acquisition, Writing – review and editing | Joanne Turner, Funding acquisition, Writing – review and editing | Yufeng Wang, Writing – review and editing | Jordi B. Torrelles, Conceptualization, Funding acquisition, Methodology, Resources, Supervision, Writing – original draft, Writing – review and editing

## DATA AVAILABILITY

All raw and processed sequencing data generated in this study was submitted to the NCBI Gene Expression Omnibus (GEO; https://www.ncbi.nlm.nih.gov/geo/) (90) under accession number GSE244851.

## ETHICS APPROVAL

Human subject studies were conducted following stringent guidelines established by the US Code of Federal and Local Regulations (OSU Institutional Review Board IRB numbers 2012H0135 and 2008H0119). Informed written consent was obtained before performing the bronchoalveolar lavage.

## ADDITIONAL FILES

The following material is available online.

### Supplemental Material

**Supplemental material (Spectrum01790-24-S0001.pdf).** Fig. S1 to S4; Legend for Tables S1 and S2.
**Table S1 (Spectrum01790-24-S0002.xlsx).** Complete list of *M.tb* DEGs.
**Table S2 (Spectrum01790-24-S0003.xlsx).** DE analysis of *M.tb* genes.

### Open Peer Review

**PEER REVIEW HISTORY (review-history.pdf).** An accounting of the reviewer comments and feedback.

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
