## [Reviewer comments · Microbiology Spectrum]

Microbiology Spectrum

Impact of the elderly lung mucosa on *Mycobacterium tuberculosis* transcriptional adaptation during infection of alveolar epithelial cells

Angélica Olmo-Fontánez, Anna Allué-Guardia, Andreu Garcia-Vilanova, Jeremy Glenn, Shu-Hua Wang, Robert Merritt, Larry Schlesinger, Joanne Turner, Yufeng Wang, and Jordi Torrelles

Corresponding Author(s): Jordi Torrelles, Texas Biomedical Research Institute

Review Timeline:

Submission Date:	July 17, 2024
Editorial Decision:	September 13, 2024
Revision Received:	September 29, 2024
Accepted:	October 10, 2024

Editor: Selvakumar Subbian

Reviewer(s): Disclosure of reviewer identity is with reference to reviewer comments included in decision letter(s). The following individuals involved in review of your submission have agreed to reveal their identity: Rajko Reljic (Reviewer #2)

Transaction Report:

DOI: <https://doi.org/10.1128/spectrum.01790-24>

Re: Spectrum01790-24 (**Impact of the elderly lung mucosa on *Mycobacterium tuberculosis* metabolic adaptation during infection of alveolar epithelial cells**)

Dear Dr. Jordi B Torrelles:

Thank you for the privilege of reviewing your work. Below you will find my comments, instructions from the Spectrum editorial office, and the reviewer comments.

Revision Guidelines

Sincerely,
Selvakumar Subbian
Editor
Microbiology Spectrum

Reviewer #2 (Comments for the Author):

In this paper, Olmo-Fontanez and colleagues investigated the impact of Mtb exposure to alveolar fluid (AF) from elderly on transcriptomic profile of the pathogen with a view to better understanding mechanisms that lead to greater susceptibility to disease in this age group. They used alveolar epithelial cells to study infectivity of pre-exposed bacteria. The results suggest that major differences could be observed between exposure to 'normal' AF and elderly AF, and this was documented by analysing expression of several genes families and pathways. These differences, they conclude, are likely at least part of the reason for

increased susceptibility of elderly people to TB.

The topic is of high interest as not much has been known about host-pathogen interplay in elderly people at risk of TB, beyond the generic 'weakened immune system' argument.

The paper is written very well, the analyses are comprehensive and the chosen molecular pathways well justified (gene families involved in respiratory burst, ESX response, PDIM, Sec secretion system etc). The statistical analyses were appropriate, and the experimental procedures well described. This paper will advance our knowledge of host-Mtb interactions in the elderly, as well as in general population.

My only minor queries are to do with the ESX secretion system, where surprisingly not many changes have been observed?

The other thing, while authors acknowledge that using pulled ALF from multiple donors is a shortcoming, it indeed could confound data and interpretations and undermine generalizability. If suppose, only one donor (out of 5) had particularly distinct composition of AF that facilitates these changes, that would weaken the argument about generalizability.

On a very minor note, I am a bit confused that 'AT' abbreviation is used for alveolar epithelial cells. They are commonly referred to as AEC, so I am not sure why AT is preferred.

Reviewer #3 (Comments for the Author):

The authors investigated transcriptional responses in *Mycobacterium tuberculosis* (M.tb) exposed to alveolar lining fluid (ALF) derived from adult (A-ALF) vs. aged (E-ALF) individuals, or alveolar epithelial cells infected with treated M.tb, and identified gene programs unique to E-ALF groups. Age-related alterations of E-ALF impact on genes endowing bacilli with cytosol escaping features and stress responses. These transcriptional changes may explain the heightened replication capacity of M.tb in host cells and perhaps the increased susceptibility of aged individuals to tuberculosis. These investigations are timely and relevant. The study is well-executed and the results are extensively discussed acknowledging also their limitations. The title should be amended as the data do not support conclusions about "metabolic adaptation", but "transcriptional adaptation" of M.tb to elderly lung mucosa.

Minor comments and few suggestions are presented below:

The variability in DEG is higher in cell-free system compared to AT conditions, about 3 times more DEG (Fig. 2C, D) vs. pretty similar DEG (Fig. 4C, D) being regulated. What is the explanation? What would be the biological significance, e.g. impact on bacilli recognition versus sub-cellular events and inflammation.

It would be beneficial to increase granularity of the presented data by providing Volcano plots for DEG for both figures 2 and 4. This will allow the reader to easily observe the level of regulation, both as a whole and specific gene (top) in case individual genes would be marked.

An additional limitation of the study may be that variability in AT is not considered. AT features may also change not only ALF, i.e. due to the biased inflammatory environment. As such interactions of aged AT with M.tb may also differ. Usage of the A549 cells does not allow evaluating such variability.

The material and method part would benefit of providing addition details. For instance, it remains unclear how M.tb was washed before processing, what buffer was used? What was the control incubation buffer for ALF? How can an effect of gentamycin on M.tb, antibiotic used in the A549 experiments, be excluded? Information on passage numbers for A549 cells should be provided.

In this paper, Olmo-Fontanez and colleagues investigated the impact of Mtb exposure to alveolar fluid (AF) from elderly on transcriptomic profile of the pathogen with a view to better understanding mechanisms that lead to greater susceptibility to disease in this age group. They used alveolar epithelial cells to study infectivity of pre-exposed bacteria. The results suggest that major differences could be observed between exposure to 'normal' AF and elderly AF, and this was documented by analysing expression of several genes families and pathways. These differences, they conclude, are likely at least part of the reason for increased susceptibility of elderly people to TB.

The topic is of high interest as not much has been known about host-pathogen interplay in elderly people at risk of TB, beyond the generic 'weakened immune system' argument.

The paper is written very well, the analyses are comprehensive and the chosen molecular pathways well justified (gene families involved in respiratory burst, ESX response, PDIM, Sec secretion system etc). The statistical analyses were appropriate, and the experimental procedures well described. This paper will advance our knowledge of host-Mtb interactions in the elderly, as well as in general population.

My only minor queries are to do with the ESX secretion system, where surprisingly not many changes have been observed?

The other thing, while authors acknowledge that using pulled ALF from multiple donors is a shortcoming, it indeed could confound data and interpretations and undermine generalizability. If suppose, only one donor (out of 5) had particularly distinct composition of AF that facilitates these changes, that would weaken the argument about generalizability.

On a very minor note, I am a bit confused that 'AT' abbreviation is used for alveolar epithelial cells. They are commonly referred to as AEC, so I am not sure why AT is preferred.

Spectrum01790-24 (Impact of the elderly lung mucosa on *Mycobacterium tuberculosis* metabolic adaptation during infection of alveolar epithelial cells)

Reviewer #1:

Response: We appreciate the reviewer's comments and thank her/him for the time and effort dedicated to review this manuscript.

Reviewer #2:

In this paper, Olmo-Fontanez and colleagues investigated the impact of Mtb exposure to alveolar fluid (AF) from elderly on transcriptomic profile of the pathogen with a view to better understanding mechanisms that lead to greater susceptibility to disease in this age group. They used alveolar epithelial cells to study infectivity of pre-exposed bacteria. The results suggest that major differences could be observed between exposure to 'normal' AF and elderly AF, and this was documented by analysing expression of several genes families and pathways. These differences, they conclude, are likely at least part of the reason for increased susceptibility of elderly people to TB. The topic is of high interest as not much has been known about host-pathogen interplay in elderly people at risk of TB, beyond the generic 'weakened immune system' argument.

The paper is written very well, the analyses are comprehensive and the chosen molecular pathways well justified (gene families involved in respiratory burst, ESX response, PDIM, Sec secretion system etc). The statistical analyses were appropriate, and the experimental procedures well described. This paper will advance our knowledge of host-Mtb interactions in the elderly, as well as in general population.

Response: We appreciate the reviewer's comments and thank him/her for the time and effort dedicated to review this manuscript.

My only minor queries are to do with the ESX secretion system, where surprisingly not many changes have been observed?

Response: We thank the reviewer for this comment. Indeed, before infection, only three ESX-4 genes and a couple of ESX-5-associated PE/PPE genes showed significant changes in expression, whereas during infection most of the observed ESX-related DEGs were downregulated. While several ESX genes have been previously associated with phagosomal rupture and *M.tb* translocation into the cytosol (e.g. *esxA*, *esxB*), it is possible that other non-ESX genes found differentially expressed here (e.g. PDIMs, oxidative stress, and virulence genes) may contribute to this process. This is mentioned in the last paragraph of the discussion (lines 447-452 in the "Clean manuscript"). Future mechanistic studies will investigate how these different bacterial determinants contribute to the increased susceptibility of the elderly to *M.tb* infection and TB pathogenesis.

The other thing, while authors acknowledge that using pulled ALF from multiple donors is a shortcoming, it indeed could confound data and interpretations and undermine generalizability. If suppose, only one donor (out of 5) had particularly distinct composition of ALF that facilitates these changes, that would weaken the argument about generalizability.

Response: We understand the reviewer's point and we agree. We have included a statement clarifying this point in the Discussion (lines 433-437 in the "Clean manuscript").

On a very minor note, I am a bit confused that 'AT' abbreviation is used for alveolar epithelial cells. They are commonly referred to as AEC, so I am not sure why AT is preferred.

Response: Since AECs is also being used by pulmonologists to define Airway Epithelial Cells, to avoid confusion and after consultation with our pulmonologist colleagues, we have been using in our papers (since 2016) the terminology Alveolar Type-I (AT-I) or Type-II (AT-II) cells, hence the ATs abbreviation to refer to Alveolar epithelial cells. We hope this clarifies the question.

Reviewer #3:

The authors investigated transcriptional responses in *Mycobacterium tuberculosis* (M.tb) exposed to alveolar lining fluid (ALF) derived from adult (A-ALF) vs. aged (E-ALF) individuals, or alveolar epithelial cells infected with treated M.tb, and identified gene programs unique to E-ALF groups. Age-related alterations of E-ALF impact on genes endowing bacilli with cytosol escaping features and stress responses. These transcriptional changes may explain the heightened replication capacity of M.tb in host cells and perhaps the increased susceptibility of aged individuals to tuberculosis. These investigations are timely and relevant. The study is well-executed and the results are extensively discussed acknowledging also their limitations. The title should be amended as the data do not support conclusions about "metabolic adaptation", but "transcriptional adaptation" of M.tb to elderly lung mucosa.

Response: We thank the reviewer for his/her time and effort dedicated to review this manuscript. As suggested, title has been changed to "*Impact of the elderly lung mucosa on *Mycobacterium tuberculosis* transcriptional adaptation during infection of alveolar epithelial cells*".

Minor comments and few suggestions are presented below:

The variability in DEG is higher in cell-free system compared to AT conditions, about 3 times more DEG (Fig. 2C, D) vs. pretty similar DEG (Fig. 4C, D) being

regulated. What is the explanation? What would be the biological significance, e.g. impact on bacilli recognition versus sub-cellular events and inflammation.

Response: Indeed, we have published that the ALF composition in the elderly is different than in adult individuals (PMID: 24584696; PMID: 35460553), with lower levels of host homeostatic hydrolytic enzymes shown to alter the cell envelope of *M.tb* (PMID: 21602490; PMID: 25748325; PMID: 28373877; PMID: 28000679). Elderly ALF also has higher levels of oxidative stress and inflammation and dysfunctional oxidized innate proteins (e.g., surfactant protein D, SP-D; PMID: 24584696), which could explain why we observed different transcriptional effects and higher variability in the number of DEGs in A-ALF- vs. E-ALF exposed *M.tb* prior to infection. These A-ALF vs. E-ALF-driven changes have an impact on *M.tb*'s recognition by the host cells and result in increased replication of E-ALF-exposed *M.tb* compared to A-ALF-*M.tb* both *in vitro* and *in vivo* (PMID: 30923818), potentially through increased translocation into the cytosol (PMID: 38185331). After ALF exposure, and once ALF-exposed *M.tb* is inside the cells, *M.tb* bacilli are exposed to the same cellular microenvironment (ATs in this case), which could start normalizing the gene expression and would explain why the number of DEGs is similar (although not identical) under both conditions (adult or elderly exposure + infection). And although the DEG numbers are similar, ~ 40% of the DEGs are specific to either adult or elderly conditions, indicating that responses during infection are different, as seen by differences in *M.tb* growth and translocation into the cytosol in the case of E-ALF (PMID: 38185331), among others. This potential explanation has been added into the discussion, (lines 293-302 in the "Clean manuscript").

It would be beneficial to increase granularity of the presented data by providing Volcano plots for DEG for both figures 2 and 4. This will allow the reader to easily observe the level of regulation, both as a whole and specific gene (top) in case individual genes would be marked.

Response: As suggested by this reviewer, we have included Volcano plots for both figures (See supplementary Figure S2).

An additional limitation of the study may be that variability in AT is not considered. AT features may also change not only ALF, i.e. due to the biased inflammatory environment. As such interactions of aged AT with *M.tb* may also differ. Usage of the A549 cells does not allow evaluating such variability.

Response: We agree with the reviewer that A549 cells do not allow us to study the variability in host ATs, which also impacts *M.tb*-host cell interactions. As the aim of this study is to understand the contribution of ALF from adult vs. elderly individuals before and during *M.tb* infection of ATs, using A549 allows us to remove the host cell variability and really focus on *M.tb* changes caused by ALF exposure. To clarify this, we have added a statement in the Discussion (lines 439-443 in the "Clean manuscript").

The material and method part would benefit of providing addition details. For instance, it remains unclear how *M.tb* was washed before processing, what buffer

was used? What was the control incubation buffer for ALF? How can an effect of gentamycin on M.tb, antibiotic used in the A549 experiments, be excluded? Information on passage numbers for A549 cells should be provided.

Response: As requested, we have added these additional details in the Materials and methods section (*lines 479-480, 484-489, 502-507 in the "Clean manuscript"*).

We thank the reviewers and the editor for their time and constructive criticism, which considerably improved this manuscript.

Re: Spectrum01790-24R1 (**Impact of the elderly lung mucosa on *Mycobacterium tuberculosis* transcriptional adaptation during infection of alveolar epithelial cells**)

Dear Dr. Jordi B Torrelles:

Your manuscript has been accepted, and I am forwarding it to the ASM production staff for publication. Your paper will first be checked to make sure all elements meet the technical requirements. ASM staff will contact you if anything needs to be revised before copyediting and production can begin. Otherwise, you will be notified when your proofs are ready to be viewed.

Sincerely,
Selvakumar Subbian
Editor
Microbiology Spectrum

Reviewer #2 (Comments for the Author):

Thank you for addressing my concerns, I am happy with your explanations

Reviewer #3 (Comments for the Author):

The authors satisfactorily addressed the comments.